# Potential of Essential Oils in the Control of *Listeria monocytogenes*

**DOI:** 10.3390/microorganisms11061364

**Published:** 2023-05-23

**Authors:** György Schneider, Anita Steinbach, Ákos Putics, Ágnes Solti-Hodován, Tamás Palkovics

**Affiliations:** 1Department of Medical Microbiology and Immunology, Medical School, University of Pécs, Szigeti St. 12, H-7624 Pécs, Hungary; anitani88@gmail.com (A.S.);; 2Central Laboratory, Aladár Petz Teaching Hospital, Vasvári Pál Street 2-4, H-9024 Győr, Hungary; akos.putics@gmail.com

**Keywords:** *Listeria monocytogenes*, essential oil, efficacy, food, method, antibacterial, preservation

## Abstract

*Listeria monocytogenes* is a foodborne pathogen, the causative agent of listeriosis. Infections typically occur through consumption of foods, such as meats, fisheries, milk, vegetables, and fruits. Today, chemical preservatives are used in foods; however, due to their effects on human health, attention is increasingly turning to natural decontamination practices. One option is the application of essential oils (EOs) with antibacterial features, since EOs are considered by many authorities as being safe. In this review, we aimed to summarize the results of recent research focusing on EOs with antilisterial activity. We review different methods via which the antilisterial effect and the antimicrobial mode of action of EOs or their compounds can be investigated. In the second part of the review, results of those studies from the last 10 years are summarized, in which EOs with antilisterial effects were applied in and on different food matrices. This section only included those studies in which EOs or their pure compounds were tested alone, without combining them with any additional physical or chemical procedure or additive. Tests were performed at different temperatures and, in certain cases, by applying different coating materials. Although certain coatings can enhance the antilisterial effect of an EO, the most effective way is to mix the EO into the food matrix. In conclusion, the application of EOs is justified in the food industry as food preservatives and could help to eliminate this zoonotic bacterium from the food chain.

## 1. *Listeria monocytogenes* as a Foodborne Pathogen

*Listeria monocytogenes* is a foodborne pathogenic bacterium, causing relatively rare symptomatic illness in the general human population; however, in high-risk groups, the disease can be serious or even life-threatening, with one of the highest fatality rates (~20%) among all the foodborne infections worldwide [1,2]. With regard to Europe, the European Center for Disease Prevention and Control (ECDC) and the European Food Safety Authority (EFSA) ranked listeriosis as the fifth most commonly reported human zoonosis with an incidence rate of 0.49 per 100,000 population. According to the latest EU One Health Zoonoses Report, 2183 invasive human cases were reported from the EU in 2021 that resulted in 923 hospitalizations and 196 deaths with a case fatality rate of 13.7% [3].

Listeriosis is a zoonotic disease, where infection primarily occurs through the consumption of contaminated food. In noninvasive listeriosis, the infection is limited to the gastrointestinal tract, is often asymptomatic or mild, and is possibly accompanied by flu-like symptoms. Colonization of the gastrointestinal tract or asymptomatic shedding, in both humans and animals has been observed [4]. In invasive listeriosis, the infection becomes disseminated, frequently causing severe illnesses such as sepsis, meningitis, pneumonia, myocarditis, or corneal ulcer. In clinical practice, this invasive form is routinely diagnosed, often needing hospitalization, while the noninvasive form remains mostly undiagnosed.

*L. monocytogenes* is a serious problem for the food industry, because bacteria can survive under many extreme conditions during food production due to their high tolerance to a wide range of environmental stresses. Unlike many other pathogens, *Listeria* can grow under refrigerated conditions, as well as at higher temperatures, ranging from −1 °C to 45 °C. Its survival is mediated by the expression of cold-shock proteins at low temperatures [5]. The flagellar motility is active under 30 °C but not at the bacterium’s optimum growth temperature of 30–37 °C [6,7]. Furthermore, the bacterium was shown to possess high tolerance for osmotic stress, high salt concentration (up to 20%), pH (4.0–9.6), and low water activities (~0.90) [8,9].

The types of food most often implicated in *Listeria* infection are ready-to-eat products that are consumed without reheating or cooking: miscellaneous fishery products, dairy products (soft cheeses and yoghurts), meat products (fermented sausages, pâtés, and cold cuts), and fruits and vegetables (salads, juices, and frozen products) [3]. In the environment, *L. monocytogenes* is ubiquitous, living in a broad range of habitats including soil, water, plants, animals, and humans. Human food can be contaminated through the entire food-processing pathway from primary production and manufacturing to final distribution. At the primary food production level (agricultural and animal farms), soil is proposed as the principal reservoir. The meat- or milk-producing animals are mostly infected via an infectious network consisting of soil, animals, feces, water, and feed (plant or silage) [10]. In manufacturing, it is commonly accepted that animals (or vegetables) to be manufactured are the initial source of contamination in food-processing facilities [11]. The eradication and the overall control of *L. monocytogenes* need complex sanitation protocols and proper monitoring because bacteria can survive under harsh conditions (low pH, low temperature, and osmotic stress), are often resistant to sanitation agents, and are capable of forming biofilms [12]. *Listeria* contamination can also occur during distribution and at the retail level where food can be contaminated from surfaces, equipment, and food-handlers; a high level of hygienic practice in the retail environment is decisive.

In the last few decades, several cases have been reported from various parts of the world [13,14,15,16]. In a comprehensive study between 1996 and 2018, 91 outbreaks and 29 recalls were identified from 27 countries [17], while, recently in the EU, eight food vehicles causing strongly evidenced outbreaks were registered in 2021 [3]. In order to minimize the number of outbreaks, continuous food monitoring must be carried out according to national and international regulations and guidelines [18]. In addition to surcharging the healthcare system, withdrawal or recall of contaminated food seriously burdens the food supply chain. According to this, the eradication of *Listeria* contamination is of key importance in the entire food production process.

## 2. Current Procedures to Combat Listeriosis

Due to the significant health risk caused by *Listeria monocytogenes*, there are regulations focusing on the handling and processing of foods at risk of *L. monocytogenes* contamination [18]. This is necessary since *L. monocytogenes* is difficult to eradicate [19,20] and can be present in different foods, such as meat, fish, milk, cheese, and fruits and vegetables [3]. In addition to personal hygiene, the crucial points of food safety regulations affect work surfaces and packaging technology. Today, the cleanliness of surfaces is assured by hydrogen peroxide and EDTA-based disinfectants [21] that are also effective in the control of biofilm formation [22]. In the case of milk, pasteurization in the recommended method, whereas, in the case of fruits and vegetables, rinsing with vinegar and water is recommended [23,24]. UV treatment and modified gas atmosphere packaging (MAP) also have the capacity to inhibit the growth of *L. monocytogenes* [25,26,27]. In general, chemical washes (chlorine and organic acids) and treatments (MAP and ozone) are effective in controlling this bacterium in foods; however, because the extensive use of chemical preservatives in foods is detrimental to human health, attention is increasingly turning to natural solutions [28]. For meat decontamination and elimination of *L. monocytogenes* from cheese during ripening, the application of bacteriophages was recently considered. Listex P100, a bacteriophage mix targeting *L. monocytogenes* was the first Food and Drug Administration (FDA)-approved bacteriophage product used in the food industry [29]. Metabolic products of certain *Lactobacillus* species, such as *L. sakei*, *L. lactis*, and their strains, during the fermentation of meats (salami and ham) were shown to hinder the proliferation and biofilm formation of this foodborne pathogen [30]. Considering ancient observations and traditional knowledge, the potential use of plant extracts has recently become the focus [31,32,33].

## 3. Essential Oils

Plant extracts are derived from leaves, buds, crops, flowers, twigs, crus, roots, and seeds. Different procedures are applied to obtain the extracts, such as cold pressing, steam and hydrodistillation, and CO_2_, supercritical fluid, and organic solvent extraction [34].

Compound compositions and their biological effects are strongly influenced by the extraction method itself [35,36] and the applied conditions [37]. EOs are concentrated hydrophobic liquids containing volatile chemical compounds from plants [38]. The raw materials typically originate from the Mediterranean and tropical regions [38].

Eos are complex mixtures. The number of identified components usually ranges from 100 to 250; however, in some oils (lavender, geranium, and rosemary) 450–500 chemicals have been found using precise instrumentation. Typically, they have 2–3 major components that are present in high concentrations (20–70%) and play an important role in biological activity [38]. Minor components are present in a low percentage. Terpenes, carbohydrates, phenols, alcohols, ethers, aldehydes, and ketones are the most studied compounds of EOs [39] and have been recognized as bioactive [40]. Terpene-containing EOs, followed by aldehyde-containing EOs (cinnamaldehyde, citral, eugenol, or thymol), are considered the most active ones, while EOs with ketones or esters (β-myrcene, α-thujone, or geranyl acetate) possess lower activities [41,42].

Many plant extracts and EOs are microbiologically active and were studied on Gram-negative and Gram-positive bacteria [43]. Due to its importance in the food industry, the number of studies dealing with the potential applications of EOs as antilisterial agents is increasing [31,44,45,46,47].

## 4. Antibacterial Mode of Action of Essential Oils

Because of its potential medical, agricultural, and industrial application, the antibacterial effect of EOs has been in the focus of several research groups. The first studies emphasized the membrane-disrupting activity of EOs, which effect was attributed to their lipophilic features [48]. Today, we already know that the picture is more complex; compounds of EOs can target (i) the cell membrane, (ii) the cell wall, (iii) energy metabolism, and (iv) genetic material [49]. Usually, the antibacterial mechanism of EOs is not a simple mode of action; rather, it is the combined result of different effects of different compounds.

It is generally known that, in comparison to Gram-negative bacteria, Gram-positive bacteria are more susceptible to EOs [50,51]. One reason may be that Gram-negative bacteria have a rigid outer membrane rich in lipopolysaccharide (LPS) which is hydrophobic, limiting the diffusion of hydrophobic EO compounds. In contrast, the thick peptidoglycan layer of Gram-positive bacteria is not dense enough to resist small antimicrobial molecules [52]. Therefore, small lipophilic compounds can easily integrate into and pass through the phospholipid bilayer, damaging the structure of cell membrane and spoiling its function [53]. Through this process, membrane permeability increases and membrane potential, a crucial factor in ATP synthesis, collapses, an alteration that is not compatible with life. Thymol and carvacrol were shown to possess such effects [54]. For undisturbed membrane functions, the presence of membrane proteins is indispensable. It was shown that certain EO compounds are able to denature membrane proteins, thereby damaging membrane functions [53,54] or affecting synthesis of the cell-wall structure [55]. Others were shown to block the function of enzymes and, thus, hinder metabolic pathways [56], bind to DNA [53], or affect protein synthesis [57]. In a recent review, these potential routes and targets on microbial cells were summarized in detail [58].

EOs act on bacterial cells in a time- and concentration-dependent manner [49]. They can be grouped on the basis of the time needed to exert their effect; we can distinguish slow- and fast-acting EO compounds and EOs. This is determined by their mode of action [59], which can be studied using different techniques.

## 5. Methods to Reveal Antilisterial Activity of Essential Oils and Their Active Components

A wide range of methods are available to study the antilisterial activities of EOs. The most effective for activity screening is the simple drop plate or paper filter-based disc diffusion method [60], which is used on the lawn of the test organism. Other studies preferred using the agar diffusion assay [61]. To define the lowest EO concentration which inhibits proliferation and which kills *L. monocytogenes*, the terms minimal inhibitory and minimal bactericidal concentration (MIC and MBC) are used. These values can be determined using macro- or microdilution methods, in glass reagent tubes [62,63,64] or in 96-well microdilution plates [65,66], respectively. Due to solubility problems of the hydrophobic EOs and their compounds, the usage of detergents (e.g., Tween-20 and Tween-80) is sometimes required [67]. Transmission and scanning electron microscopy (TEM and SEM) are adequate techniques to visualize the morphological changes accompanying antimicrobial effects [31,47,62,68,69]. To identify active compounds responsible for the antilisterial activity of an EO, bioautography is a proper method. This is based on thin-layer chromatography (TLC) performed on silica gel [70]. Active compounds are identified on the basis of their Rf values, while nonidentifiable active volatile compounds can be cut out from the silica gel and analyzed using the headspace solid-phase microextraction method coupled to gas chromatography–mass spectrometry (HS-SPME/GC-MS) [71]. With this, the purity or percentage composition of antimicrobial active compounds from silica gel can be determined.

For the biofilm-inhibitory or biofilm-degrading capacity of EOs or active compounds, the classical crystal violet staining assay, performed in 96-well microplate format, is most commonly used [66,72,73,74]; however, other methods, such as the TEMPO system (bioMérieux), VIDAS system (bioMérieux), or the discrete element method (DEM), have also been suggested [75]. Polystyrene, polypropylene, polyethylene, glass, and stainless steel are typically tested abiotic surface materials [31,45,72]. SEM and Confocal laser scanning electron microscopy (CLSM) are used to analyze changes in the biofilm integrity as a result of treatment [76].

The molecular changes accompanying the antilisterial activities of EOs and their compounds uncovered using the above methods can be further analyzed with molecular biological tools. Changes in protein profiles are often studied with 1D [47,62] or with the more detailed 2D polyacrylamide gel electrophoresis (PAGE). A fast alternative of 1D PAGE is capillary electrophoresis, in which expression differences between treated and control samples can be detected in a couple of minutes. A similar quantitative analysis uses liquid chromatography–mass spectrometry (LC–MS/MS) [69].

Further analysis of affected proteins, isolated with 2D PAGE, requires cutting and extraction from the acrylamide gel, followed by separation with liquid chromatography–mass spectrometry (LC–MS/MS) [69,77].

Today, high-throughput molecular biological methods are effectively used to uncover the underlying molecular events of bacterial metabolism in the presence of EOs or their active compounds. Whole-transcriptome analysis (WTA) is a proper approach to get a global view of the level of RNA synthesis [78], while the involvement of individual target genes, such as virulence-associated genes, can be further analyzed more precisely using the reverse transcription quantitative polymerase chain reaction (RT-qPCR) [47,79,80].

Certain enzymatic assays are also preferably used to gain insight into which part of the metabolism is affected on an enzymatic level. Measurement of the level of β-galactosidase and ATPase gives feedback about the energy metabolism of *L. monocytogenes*, while the appearance of alkaline phosphatase outside the cell suggests weakened cell-wall integrity [47,62]. Membrane integrity can also be studied by quantifying the appearance of extracellular DNA in the medium [80].

## 6. Essential Oils with Antilisterial Activities

A broad range of EOs have been investigated for their antilisterial activity in the last two decades. Results of these tests are summarized in Table 1. Most of the represented articles focused on the activity of single EOs, but some also investigated synergistic effects when different EOs were combined [81,82]. The significance of this is that most EOs have a strain-dependent effect on *L. monocytogenes* [44,82]; therefore, a combination of different EOs could be an adequate approach against this foodborne pathogen in practice.

Results indicate that members of the Lamiaceae family, involving different *Thymus* and *Oregano* spp., showed the most extended antilisterial activity. Additionally, *Cinnamomun* spp. was proven to be effective. Typically, the major compounds were responsible for the antilisterial activities, especially if they belonged to the groups of mono- and sesquiterpenes. For the antilisterial effects, in most cases, compounds such as carvacrol, thymol, p-cymene, alpha-pinene, terpinene, or citral were responsible [44,83].

Most of the investigated EOs were extracted by steam distillation and originated from different countries and different producers. Especially in earlier studies, the compound composition (determined by gas chromatography) of the investigated EOs was not presented, which is a shortcoming that compromises the comparability of the different studies. This is an important issue as the compound composition of EOs is determined by geographical localization, weather, and time of harvest [84,85], which can influence the test results. A good example is that the compound composition of oregano EO in three studies differed significantly [44,86,87]; in the study of Gottardo, carvacrol content was 91%, whereas, in the studies of Maggio and Pesavento, these values were 68% and 71.8%, respectively. This was also the case with thyme showing differences in major compound content, albeit without influencing the antilisterial activity [44,45,88].

Another problem in the comparability of the results is that the bacterial cell numbers applied in different studies, either for the simple drop plate method or for MIC and MBC determinations, showed discrepancies. Furthermore, during tests, different media were used, e.g., Luria–Bertani (LB), Mueller–Hinton Broth (MHB), Brain Heart Infusion (BHI), and Peptone Yeast glucose (PYG) [60,62,63]. In most cases, in vitro tests were performed at 37 °C, whereas tests were only rarely conducted at lower temperatures under refrigerated conditions, which are mostly applied in food systems. Certainly, it would be a mistake to overstate the importance of the above factors. Moreover, since growth conditions influence gene regulation and, thus, phenotypic heterogeneity in bacteria [89], these factors could also influence the sensitivity of *L. monocytogenes* to EOs.

The antilisterial effect of EOs, summarized in Table 1, was mostly investigated using the standard disc diffusion technique [90], as this is the simplest screening method. In positive cases, the diameter of an inhibition zone around the EO spot or disc was typically between 20 and 30 mm, as demonstrated in several cases: black seed oil (31.50 mm) [90], broccoli sprout extract (17.84 ± 0.34 mm) [91], *Citrus medica* L. var. *sarcodactylis* Swingle *citron* oil (23.45 ± 1.23 mm) [78], *Ceratonia siliqua* EO (17 ± 0.3 mm) [65], *Hibiscus surattensis* L. calyces EO (25.26  ±  1.53 mm) [63], and Melaleuca *alternifolia* EO (30 ± 8.8 mm). In addition to this method, the agar diffusion assay [44,72,78], disc volatilization method [92], and plate colony counting [62] in pure, nanoliposome [91], nanocapsule [91], nanoemulsion [60,93,94], and liposome [95] systems were used. The advantage of the use of nanoemulsions is that this formulation is able to increase the biological activity and stability of EOs [96]. Nanoemulsion systems consist of three components: EO, water, and a nonionic surfactant, e.g., Tween-80 [60,94]. These three components are mixed, and then the particle size can be decreased using a sonicator [60].

Considering the tests, we have to emphasize again that the effect of EOs can be strain-dependent. A good example demonstrating this was a recent study in which the EO of *Melissa officinalis* was tested on three strains of *L. monocytogenes* (LMG 13305, 16779, and 16780), and three different inhibition zone diameters were found: 38.4 ± 4.2 mm, 54.6 ± 1.3, mm and 48.6 ± 1.7 mm, respectively [74]. In the case of *Schinus terebinthifolius* Raddi, EOs were produced from both ripe and unripe fruits. The inhibition zone of the latter was 35.22 ± 0.79 mm, while that of the former was 40.86 ± 0.31 mm [97].

Another aspect to be considered is which part of the plant and which procedure were used for the extraction. In a recent study, the authors revealed that the antilisterial effect of thyme was the strongest if acetone extract from the leaves was used, while ethanolic extract from the seeds exhibited the lowest antilisterial activity [98].

In most of the recent studies, kinetic assays were also performed in order to reveal the course of the antilisterial effect [68,99,100]. Kinetic curves are necessary if molecular changes on genomic and proteomic levels, accompanying the antilisterial effect, are intended to be investigated [78]. Through these analyses, the antibacterial mode of action of certain EOs can be revealed.

Since *L. monocytogenes* is able to form biofilms, a number of experiments focused on the antibiofilm capacity of EOs [45,67]. In such experiments, the biofilm-forming capacities of *L. monocytogenes* strains were hindered by *Cinnamomun zeylanicum* or *Eugenia caryophyllata* EOs [101]. Furthermore, it was also investigated whether EOs have the ability to destroy the already established biofilm on certain surfaces. They found that, within 2 h, clove could already drastically weaken the established biofilm. Such capacity is not typical for all EOs, as demonstrated by Guo et al. After establishing a firm biofilm in 72 h, they treated it with *Citrus Changshan-huyou* EO for 24 h, but the formed biofilm remained intact after treatment [31].

Revealing the antilisterial effect in in vitro studies is inevitable before considering practical uses; from this perspective, the results of screening and exploring the antimicrobial mode of action are all relevant issues, but the real challenge is always how a certain EO with potential antilisterial effects performs under harsh conditions if applied in different food systems.
microorganisms-11-01364-t001_Table 1Table 1Summary of antilisterial effects of different EOs.Source of Essential OilMajor Compounds Investigated in the StudyApplied MethodExperimental ConditionMIC (μL/mL) orInhibition Zone (mm)MBC (μL/mL)Reference*Achillea millefolium*Caryophyllene, 1,8-cineole, bornyl acetate, 1-terpinen-4-ol, β-pinene, camphorAgar disc diffusion assay, MIC, MBC, biofilm assayBHI broth using a broth microdilution method in the 96-well round-bottomed polystyrene microtiter plates31.3 μL/mL16 mm62.5 μL/mLJadhav, Shah et al. [72]*Allium sativum* L.
Disc diffusion method, MIC,96-well micro-dilution plates with U-bottom wells37.5 μL/mL
Razavi Rohani, Moradi et al. [102]*Allium vineale*
Disc diffusion method, diameter of inhibition zoneTurkish Herby Cheese8–15 mm, depending on the strain
Sagun, Durmaz et al. [103]*Brassica oleracea* var. *italica*
Nanoliposome, nanocapsule, AOA, SEM, MICBSE nanoliposome, ricotta cheese0.8 μL/mL
Azarashkan, Farahani et al. [91]*Caryophyllorum salisque*
Nanoemulsion, MIC, characterization of NEs, agar well diffusion method, inhibition zone, TEMEgyptian Talaga cheese, 96-well plate45.2 ± 34.25 mm
Elsherif and Talaat Al Shrief [96]*Ceratonia siliqua*Nonadecane, heneicosane, naphthalene, 1,2-benzenedicarboxylic acid dibutylester, heptadecane, hexadecanoic acid, octadecanoic acid, 1,2-benzenedicarboxylic acid, phenyl ethyl tiglate, eicosene, farnesol 3, camphor, nerolidol, *n*-eicosaneAgar diffusion methodMIC, MFC, MTT test, cytotoxicity assay96-well microplates2.5 μL/mL17 ± 0.3 mm
Hsouna, Trigui et al. [65]*Chaerophyllum macropodum*
Disc diffusion method, diameter of inhibition zoneTurkish Herby cheese7–13 mm, depending on the strain
Sagun, Durmaz et al. [103]*Cinnamomum zeylanicum*
Agar disc diffusion assayMIC, MBC, sensory evaluationRaw minced meat7.5 μL/mL7–28.7 mm, depending on strain and concentration7.5 μL/mLPesavento, Calonico et al. [44]*Cinnamomum cassia*Cinnamaldehyde, 2-propenal, acrylic acid, benzaldehydeAgar diffusion, MIC, EO microencapsulation, sensory analysis, encapsulation efficiencyItalian salami3 μL/mL0.8–38 mm, depending on the concentration
Gottardo, Biduski et al. [86]*Cinnamomum cassia Blume**trans*-Cinnamaldehyde, cinnamyl acetateMIC, biofilm, DEM method96-well microtiter plates0.41 ± 0.02 μL/mL
Bermúdez-Capdevila, Cervantes-Huamán et al. [67]*Cinnamomum zeylanicum*Eugenol, cinnamaldehyde, cinnamyl acetate, β-phelandreneMIC, biofilm, DEM method96-well microtiter plates4.56 ± 0.2 μL/mL
Bermúdez-Capdevila, Cervantes-Huamán et al. [67]*Cinnamomun zeylanicum*CinnamaldehydeMIC, biofilm, SEM, Box–Behnken experimental designPlate, Falcon tubes1.6 μL/mL
Vidács, Kerekes et al. [45]*Cinnamomun zeylanicum*
MIC, MBC, biofilm, eDNS, cytotoxicity, qPCR24-well culture platerange 10 μL/mL50 μL/mLBanerji, Mahamune et al. [80]*Citrus Changshan-huyou*
Disc diffusion assay, MIC, time-to-kill assay, SEM, TEM, RNA-seq, biofilm assay, SEM, CLSMRibo-Zero rRNA Removal Kit40 μL/mL25.48 ± 1.41 mm80 μL/mLGuo, Gao et al. [31]*Citrus medica* L. var. *sarcodactylis* Swingle
Agar diffusion assay(MIC)96-well tissue culture plate40 μL/mL/23.45 ± 1.23 mm
Guo, Hu et al. [78]*Citrus sinensis*
Nanoemulsions(MIC, MBC),disc diffusion assay, time-to-kill assay, antibiofilm assayMHA, 96 well microtiter plate, 24 well microtiter plate9 ± 0.31 mm–13 ± 0.75 mm,depending on the concentrations
Das, Vishakha et al. [60]*Citrus limon* var. *pompia*
Disc volatilization method, time-to-kill assay, SEM, TEMRicotta salata cheese0.086 μL/mL
Fancello, Petretto et al. [92]*Citrus limon*
var.*prompia*
Limonene, γ-terpinene, α-terpineol, β-pinene, β-myrcene, citralShelf-life evaluation, cell constituent release, crystal violet assay, SEM, sensory evaluationsRTE vegetable salads (carrot, tomato, green oak lettuce, red cabbage)

Parichanon, Sattayakhom et al. [104]*Cuminum cyminum*
Nanoemulsion, MIC, characterization of NEs, agar well diffusion method, inhibition zone, TEMEgyptian Talaga cheese, 96-well plate50.23 ± 15.7 mm
Elsherif and Talaat Al Shrief [96]*Cymbopogon citratus*
Liposome systemCheese

Cui, Wu et al. [95]*Cymbopogon citratus*Geranial, neral, limonene, geraniol, geranyl acetateGene expression assayPureLink RNA Mini Kit

Hadjilouka, Mavrogiannis et al. [105]*Eugenia* spp.Eugenol, eugnyl acetate, caryophylleneMIC, biofilm, DEM method96-well microtiter plates0.2 ± 0.02 μL/mL
Bermúdez-Capdevila, Cervantes-Huamán et al. [67]*Eugenia caryophyllata*
Disc diffusion method, MICBeef hot dogs15.6–31.2 μL/mL,depending on the strain
Singh, Singh et al. [83]*Eugenia caryophyllata*
MIC, MBC, biofilm, eDNS, cytotoxicity, qPCR24-well culture plate,1.5 μL/mL
Banerji, Mahamune et al. [80]*Hibiscus surratensis* L. calyceβ-Caryophyllene, menthol, methyl salicylate, camphor, germacrene DDisc diffusion method, MIC, MBCBroth0.15 ± 0.05 μL/mL25.26 ± 1.53 mm0.083 ± 0.04 μL/mLAkarca [63]*Laurus nobilis*
Liposome-coated, MIC, MBC, SEMSilver carp (*Hypophthalmicchthys molitrix*)45 μL/mL50 μL/mLAala, Ahmadi et al. [106]*Melaleuca alternifolia*Terpinen-4-ol, gamma-terpinene, α-terpinene, α- terpineol, terpinolene, α-pineneAgar disc diffusion method, MIC, death–time curve, SEMGround beef, 96-well microplates0.10 μL/mL30 ± 8.8 mm0.15 μL/mLSilva, Figueiredo et al. [46]*Melissa officinalis*β-caryophyllene, *cis*-1,2-dihydroperillaldehyde, caryophyllene oxide, geranyl acetate, citronellal, β-citronellol, photocitral A, (*E*)-methyl geranate, β-linaloolAgar disc diffusion assay, MIC, time-kill curves determination, quorum sensingWatermelon0.5 μL/mL 38.4 ± 4.2–54.6 ± 1.3 mm, depending on the strain
Carvalho, Coimbra et al. [74]*Mentha piperita*Pulagone, isomenthone, piperitenone, menthone, piperitoneHigh-pressure processing, separatelyAyran (yoghurt)

Evrendilek and Balasubramaniam [107]*Moringa oleifera*Palmitic acid, phytol, ethyl palmitateMIC, double dilution method, virulence gene activity, time-to-kill curve, LSCM analysisMozzarella cheese, cheddar cheese, parmesan cheese, camembert cheese10 μL/mL
Cui, Li et al. [47]*Moringa oleifera*Palmitic acid, phytol, ethyl palmitate, hexadecanalMIC, MBC, moringa–chitosan nanoparticles, FTIR, SEM, AFM, color and sensory evaluation, time-to-kill curveFresh hard cheese (cheddar)10 μL/mL10 μL/mLLin, Gu et al. [64]*Nigella sativa*Carvacrol, thymol, thymohydroquinone, thymoquinone, limonene, carvone, *p*-cymene, γ-terpineneStandard disc diffusion techniqueAm1 plate28.2 ± 2.0–39.5 ± 1.1 mm, depending on the strain
Nair, Vasudevan et al. [90]*Origanum majorana*Terpinene-4-ol, γ-terpinene, β-phellandreneMIC, biofilm, SEM, Box–Behnken experimental designPlate, Falcon tubes6.3 μL/mL
Vidács, Kerekes et al. [45]*Origanum vulgare*Monoterpene, carvacrol, p-cymene, sesquiterpenesAgar disc diffusion assayMIC, MBC, sensory evaluationRaw minced meat0.062–0.12 μL/mL, depending on the strain10.3–27.3, depending on strain and concentration0.062–0.12 μL/mL, depending on the strainPesavento, Calonico et al. [44]*Origanum vulgare*Carvacrol, *o*-cymene, thymolPetri dish, confocal laser scanning, MIC,GEN III microplates2.50 μL/mL
Maggio, Rossi et al. [87]*Origanum*Carvacrol, linalool, p-cymeneThermal inactivation by sous-vide processing, D and z-valuesAtlantic salmon (*Salmo salar)*

Dogruyol, Mol et al. [108]*Origanum vulgare*Thymol, carvacrolInoculation of chicken fillets, sensory evaluation, changes in shelf-life studyFresh chicken breast meat fillets

Khanjari, Karabagias et al. [109]*Origanum vulgare*Carvacrol, p-cymene, caryophyllene, terpineneAgar diffusion, MIC, EO microencapsulation, sensory analysisItalian salami3 μL/mL0.8–38 mm, depending on the concentration
Gottardo, Biduski et al. [86]*Origanum vulgare* subsp. *hirtum*α-thujene, p-cymene, gamma-terpinene, thymol, carvacrolSpreading, sensory evaluationFeta cheese

Govaris, Botsoglou et al. [25]*Origanum vulgare* L.CarvacrolCarvacrol encapsulation on chia mucilage nanoparticle (CMNP) and flaxseed mucilage nanoparticle (FMNP)Carvacrol was encapsulated in mucilage (chia and flaxseed) using the BIC, time-to-kill assay96-well microplate

Cacciatore, Maders et al. [99]*Phoenix dactylifera* L.3,4-dimethoxytoluene, 5,9-undecadien-2-one,9-octadecenoic acid,2,6-dimethoxytolueneInhibition zones, inhibition activity, agar well diffusion assayChicken meat13 mm
Al-Zoreky and Al-Taher [110]*Picea excelsa*β-Pinene, α-pinene, limonene, camphene, delta-3-carene, β phellandrene, 1,8-cineole, traces of sabinene, α-terpineol, terpinen-4-olMIC, LBC, MBCBroth, plate0.15 ± 0.02–0.67 ± 0.26 μL/mL2–6 μL/mLCanillac and Mourey [111]*Picea excelsa*β-Pinene, α-pinene, limonene, campheneMIC, MBC, simplified method, kinetic studiesCheese0.25–0.26 μL/mL2–2.1 μL/mLCanillac and Mourey [112]*Pimenta dioica*
Disc diffusion method, MICBeef hot dogs15.6–31.2 μL/mL,depending on the strain
Singh, Singh et al. [83]*Plectranthus amboinicus* (Lour.) Spreng.Thymol, p-cymene, β-myrcene, α-terpinoleneMIC, MBC, time-to-kill assay, bacterial anti-adhesion assayBeef patties2 μL/mL4 μL/mLDutra da Silva, Bernardes et al. [73]*Prangos ferulacea*
Disc diffusion method, diameter of inhibition zoneTurkish Herby cheese8–13 mm, depending on the strain
Sagun, Durmaz et al. [103]*Prunus armeniaca*Benzaldehyde, benzoic acid, mandelonitrileChitosan films, sensory evaluationSpiced beef

Wang, Dong et al. [113]*Rosmarinus officinalis*1,8-cineole, α-pinene, sesquiterpenesAgar disc diffusion assay,MIC, MBC, sensory evaluationRaw minced meat5–30 μL/mL, depending on the strain6–19.7 mm, depending on strain and concentration5–30 μL/mL, depending on the strainPesavento, Calonico et al. [44]*Rosmarinus officinalis*
Disc diffusion method, MICBeef hot dogs62.5–125.0 μL/mL
Singh, Singh et al. [83]*Syzygium aromaticum*
Petri dishChicken frankfurters

Mytle, Anderson et al. [114]*Syzygium aromaticum*
MIC, MBC, plate colony counting, time-to-kill analysis, TEMPYG liquid medium0.5 μL/mL1 μL/mLCui, Zhang et al. [62]*Salvia officinalis*
Agar disc diffusion assayMIC, MBC, sensory evaluationRaw minced meat60 μL/mL6–15.7 mm, depending on strain and concentration60 μL/mLPesavento, Calonico et al. [44]*Salvia officinalis*
Disc diffusion method, MICBeef hot dogs125.0–250.0 μL/mL
Singh, Singh et al. [83]*Salvia rosmarius*
Liposome-coated, MIC, MBC, SEMSilver carp (*Hypophthalmicchthys molitrix*)5 μL/mL10 μL/mLAala, Ahmadi et al. [106]*Salvia officinalis* L.β-Pinene, camphor, β-thujene, 1.8-cineole, α-humulene, endoborneolSous-vide cook-chill (SVCC), MIC*Maronesa* male bovines31.25 μL/mL
Moura-Alves, Gouveia et al. [115]*Satureja horvatii**p*-Cymene, thymol, thymol methyl ether, γ-terpinene, α-pinene, α-terpineneMIC, MBC, MYC, modified micro-dilution technique, sensory evaluation96-wells microplates, pork meat medium0.57 ± 0.03 μL/mL1.15 ± 0.01 μL/mL)Bukvicki, Stojkovic et al. [116]*Schinus terebinthifolius* RaddiMonoterpenes such as α-pinene, β-Pinene, myrcene, limonene, *D*-germacreneDisc diffusion method, MIC, MBC, inhibition zoneCheese6.799–6.820 μL/mL35.22 ± 0.79–40.86 ± 0.31 mm6820–13.598 μL/mLda Silva Dannenberg, Funck et al. [97]*Tetraastris catuaba*β-Caryophyllene, α-copaeno, α-himachalene, *iso*-sylvestrene, linalool butanoate, α-pinene, guaieneNanoemulsion, thermal analysis, stability test, TEM, biofilm assay, SEMMicrotiter plate, BHI agar

Silva, de Souza Arruda et al. [93]*Trachyspermum ammi*Thymol, p-cymene, γ-terpinene, α-terpinene, α-thujeneNanoemulsions, MIC, MBCTurkey fillet preparation8 μL/mL
Kazemeini, Azizian et al. [94]*Thymbra capitata*
MIC, MBC, WGS, antibiotic susceptibility test, *Thymbra capitata* evolution asaySkimmed milk0.15–0.30 μL/mL,depending on the strain0.20–0.40μL/mL,depending on the strainBerdejo, Pagan et al. [117]*Thymus capitatus*Carvacrol, p-cymeneMIC, biofilm, DEM method96-well microtiter plates2.56 ±0.17 μL/mL
Bermúdez-Capdevila, Cervantes-Huamán et al. [67]*Thymus eriocalyx*Thymol, α-phellandrene, *cis*-sabinene hydroxide, 1,8-cineole, α-pineneDisc diffusion method, MIC, bactericidal kinetics, TEMBHI, MH0.25 μL/mL19–44 mm,depending on the concentration
Rasooli, Rezaei et al. [68]*Thymus x-porlock*Thymol, α-phellandrene, *cis*-sabinene hydroxide, 1,8-cineole, α-pineneDisc diffusion method, MIC, bactericidal kinetics, TEMBHI, MH0.25 μL/mL/19–40 mm, depending on the concentration
Rasooli, Rezaei et al. [68]*Thymus vulgaris*Monoterpenes and sesquiterpenes, p-cymene, thymolAgar disc diffusion assayMIC, MBC, sensory evaluationRaw minced meat0.25 μL/mL11–33.5 mm, depending on the strain and concentration0.25 μL/mLPesavento, Calonico et al. [44]*Thymus vulgaris*p-Cymene, γ-terpinene, thymol, carvacrol, β-bisaboleneSpreading, sensory evaluationFeta cheese

Govaris, Botsoglou et al. [25]*Thymus vulgaris*
Disc diffusion method, MICBeef hot dogs7.8–15.6 μL/mL,depending on the strain
Singh, Singh et al. [83]*Thymus vulgaris*γ-Terpinene, thymol, p-cymeneMIC, biofilm, SEM, Box–Behnken experimental design
6.3 μL/mL
Vidács, Kerekes et al. [45]*Thymus vulgaris* L.CarvacrolCarvacrol encapsulation on chia mucilage nanoparticle (CMNP) and flaxseed mucilage nanoparticle (FMNP),carvacrol was encapsulated in mucilage (chia and flaxseed) using the BIC, time-to-kill assay96-well microplate

Cacciatore, Maders et al. [99]*Thymus vulgaris* L.α-Pinene, p-cymene, thymol, linalool, γ-TerpineneMICCheese, microtiter plate2.5 μL/mL
de Carvalho, de Souza et al. [88]*Thymus zygis*
Disc diffusion method, vapor-phase antimicrobial activity determination, MIC, time-to-kill curves, motility assay, biofilmChicken juice, lettuce leaf model, ZHT-treated skim, milk, spinach0.5 μL/mL41.55 ± 2.63–55.04 ± 3.64 mm, depending on the strain
Coimbra, Carvalho et al. [100]*Zataria multiflora Boiss*α-Pinene, p-cymene, α-terpinene, eucalyptol, α-terpineolSensory analysis, MBC, gene expression assay (RNS extraction, purification, RT-PCR)Broth and minced rainbow trout0.31–0.9 μL/mL,depending on the temperature0.625–1.25 μL/mL,depending on temperaturePilevar, Hosseini et al. [79]*Ziziphora clinopodioides*Carvacrol, thymol, *p*-cymene, *ɣ*-terpineneChitosan–gelatin film, sensory evaluationMinced rainbow trout

Kakaei and Shahbazi [118]


## 7. Essential Oils, Modulating Virulence, and Biofilm Formation of *L. monocytogenes*

*L. monocytogenes* has a wide variety of virulence factors which contribute to pathogenicity and enable switching between saprophytism and virulence, depending on the environmental setting [119]. Toxins are major determinants of the virulence in *Listeria* including hemolysins (listeriolysin O), phospholipases, and the toxin/antitoxin MazEF, while further factors are secreted, surface-associated or intracellular proteins, such as internalins, siderophores, cold-shock proteins, and the ActA protein. Since EOs have been shown to have modulating effects on the expression of virulence factors [47,79], their use can be an effective strategy against *L. monocytogenes* in prevention or control in food-related environments.

Listeriolysin O mediates the lysis of the phagosome in the infected host cell, enabling the escape of bacteria into cytosol, where they can replicate, while phospholipases lyse the membrane of endocytic and secondary vacuoles. Bay, clove, cinnamon, nutmeg, and thyme EOs were shown to significantly reduce the production of listeriolysin O, while the EO of clove diminished phosphatidylcholine-specific phospholipase C activity of *L. monocytogenes* [120]. The treatment of tea tree [121] or zedoary turmeric EOs was shown to inhibit secretion (and transcription) of listeriolysin O (hly) and invasion-associated protein p60 (iap) [122].

*Listeria monocytogenes* is a flagellated bacterium with a temperature-dependent motility; the bacterium is motile at 30 °C, but loses this ability at 37 °C [6,123]. Genome-wide transcriptome analysis suggested that exposure of *L. monocytogenes* to fingered citrone (*Citrus medica* L. var. *sacrodactylis* Swingle) EO increased motility [123], while proteomic analysis revealed that thyme EO may inhibit flagellar motility and synthesis of the flagellum, inducing structural damage in flagellar filaments [69]. Furthermore, EO of *Cannabis sativa* L. showed a significant inhibitory effect; it downregulated flagellar motility genes and the positive regulatory factor A (prfA), which is the regulator of the central virulence gene cluster in *L. monocytogenes* [124]. *L. monocytogenes* is able to form biofilms [125]. Garlic, onion, and cinnamon EOs showed an effective antibiofilm activity against *L. monocytogenes* [126]. *Cannabis sativa* L. EO treatment caused a significant reduction in biofilm formation and invasion [124]. Lavender EO was found to also have antibiofilm activity on *L. monocytogenes*, which varied in terms of biofilm development at different temperatures [127]. Further antibiofilm activities have been demonstrated in the case of several other EOs of *Thymus zygis* subsp. *gracilis* [128], *Melissa officinalis* [74], *Plectranthus barbatus* [129], *Pimenta dioica* (eugenol) [130], and *Eucalyptus* species [131].

Focusing on special food-contact surfaces, EOs of *Origanum hirtum* and *Corydothymus capitatus*, and the hydrolate of *Citrus aurantium* were demonstrated to control *Listeria* biofilms, particularly preventing biofilm formation on stainless-steel or polystyrene surfaces [132]. A reduction in *L. monocytogenes* biofilms on stainless-steel or polystyrene surfaces was also demonstrated using the EOs of thyme, oregano, carvacol [133], or *Satureja thymbra* [131]. A further interesting antibiofilm strategy might be the incorporation of EO constituents into artificial copolymer surfaces, as shown for carvacrol and cinnamaldehyde. This surface-covering film, containing the EO constituents, had a significant bacteriostatic and antibiofilm effect against *L. monocytogenes* [134].

Considering permeability issues in biofilms, the formulation of the antibiofilm agents may be of key importance in their antimicrobial effect. The nanoemulsion of *Carum copticum* EO produced by low-energy sonication was shown to have a higher antibacterial efficiency than its nonencapsulated version [135]. Light-controllable chitosan micelles loaded with thymol effectively eradicated *L. monocytogenes* biofilms [136]. Moreover, the encapsulation of the EO compounds carvacol and eugenol in a micellar nonionic surfactant solution was tested against *L. monocytogenes* in growing colony biofilms [137].

In terms of species composition, biofilms can be heterogeneous if they are built up by more bacterium species. The interactions between bacterial populations can contribute to increased resistance. As shown in a study examining dual-species biofilms formed by *L. monocytogenes* along with an additional bacterium species, higher EO concentrations of cinnamon, marjoram, and thyme were required to eliminate living cells from the matrix if biofilms were heterogeneous [138].

Combining EOs (or their components) can be a potential strategy to enhance their antibiofilm activity. Thymol and cinnamaldehyde elicited a synergestic effect with streptomycin against *L. monocytogenes*, which was sufficient to eradicate biofilms formed by this bacterium [139].

## 8. Essential Oil-Based Control of *Listeria monocytogenes* in Food

Early demonstrations of the in vitro efficacy of several EOs or EO compounds on *L. monocytogenes* suggested their potential use in food industry. The challenge is big since, due to its high fatality rate, listeriosis is considered one of the most common causes of death worldwide among foodborne illnesses [140]. Since different meats, milk, fruits, and vegetables can all be sources of infection [3,141,142], the number of publications in which the role of EOs as food preservatives was tested and demonstrated is increasing. Furthermore, several studies investigated the application of EO-based combined methods for food preservation, such as adding nisin, nanomaterials, etc., but these procedures are outside of the scope of this review, in which we strictly focus on findings in which EOs or some of their compounds were tested and applied in pure, nanoemulsion, and encapsulated forms, either alone or together with a carrier material. Most of the studies were carried out at low temperatures (4–8 °C), but there were some studies in which the antilisterial efficacies of EOs were compared at different temperatures, typically at 4 and 25 °C [116,143], as well as between two lower temperatures such as 2 and 8 °C [108,144,145].

### 8.1. Investigated Food Types

Concerning investigated food types, one common category was meat, which is rich in nutrients strongly supporting the proliferation of bacteria [146]. A significant number of recently published articles focused on the clearance of *L. monocytogenes* from meats, such as beef [64,93,147], poultry [148] pork [149], and sausage. Furthermore, meats of animal species corresponding to geographical locations, such as camel [150] and black wildebeest [151] were also investigated. Since fish is of primary importance in the world’s protein supply and it is frequently contaminated with *L. monocytogenes*, disinfection practices were investigated in or on different fish species, such as salmon [108,152], mackerel, flounder, carp, catfish [153], and other fish and fishery products [154], including shrimp [155].

Cheeses, particularly soft cheeses, have been implicated in listeriosis outbreaks worldwide [142]. Therefore, several studies focused on EO-based practices that influence the survival of this foodborne pathogenic bacterium in and on cheeses [64,91,96,97]. In a recent study, milk, another basic compound in the food industry used in ice creams and puddings, was investigated [156].

Possible fecal contamination in crop production justifies the application of disinfection practices on vegetables against *L. monocytogenes*. In a previous review, the disinfection capacity of EOs against different pathogenic bacteria, partially *L. monocytogenes*, were summarized [157]. Since that review, the antilisterial activity of EOs against *L. monocytogenes* has been revealed in fresh-cut vegetable systems [158,159], salads [104,160,161], and frozen vegetables, such as broccoli, carrot, pea, cauliflower, spinach, beans, and their mixtures [98]. Decontamination practices of fruits such as apple [162,163], melon, papaya [164], cantaloupe [69], pineapple, mango [158,165], watermelon [158,166], strawberry [167], and tomato are also relevant, because of their disposition to possible *L. monocytogenes* contamination [141], whether on intact or on fresh-cut surfaces of fruits. 

In addition to raw fruits and vegetables, their liquid products were targets of research. The antilisterial efficacy of EOs was tested in cucumber [168], pineapple [158], and mango juices [165], as well as in soymilk [169].

### 8.2. Antilisterial Tests of EOs Performed in Different Foods

Early in vitro tests, typically based on results of the drop plate method, revealed that thyme, oregano, and clove were among the most effective EOs on *L. monocytogenes* [170,171]. Consistent with these observations, thyme was the most studied compound in vitro or as an EO in different food systems [69,98,147,158,159,172] (Table 2). Scollard and coworkers confirmed the antilisterial activity of thyme on the surface of fresh-cut lettuce and melon [158]. In their system, the effect of undiluted, sprayed, or dipped thyme was very fast, reducing the number of *L. monocytogenes* CFU below the detection limit in 1 day. A similarly fast effect of thyme was observed when applied on frozen vegetable samples, naturally contaminated with *L. monocytogenes* [98]. The authors showed that the antilisterial effect depended on the extraction method, as acetone extract from the leaves was the strongest, while ethanolic extract from the seeds exhibited the lowest antilisterial activity. A 0.72 log reduction in CFU of *L. monocytogenes* was observed after 1 day when an alginate-based coating was applied with 0.65% thyme concentration [69]. The CFU difference between control and treated samples became nearly 2.5 log on day 16 of incubation. A similar result was obtained when a chitosan-starch film with 1% or 2% thyme content was applied on the surface of beef. In this case, the starting CFU of the meat slice was 5 log CFU/g, increasing to 8.8 log CFU/g under treatment, compared to the control, where this value reached 10 log CFU/g during the 21 day incubation period [147]. This and other studies demonstrated that, in several cases, only an inhibitory effect could be achieved, not an antibacterial one, which would have been ideal.

For a drastic antibacterial effect, the application of EOs in a gaseous phase can be the proper approach in the case of certain food types. Both thyme and oregano (the latter being the most often studied EO against *L. monocytogenes*) could reach a 2.1 log CFU/g drop within 24 h incubation if applied in gaseous form. Using this method, the EO treatment left a nearly 4 log CFU/g bacteria on the surface of radish sprouts [159]. Similar to this, a decrease of 2 log from 4.5 log CFU/g was measured when oregano (2% or 4%) was applied in a mixed chitosan film on the surface of pork. This effect was stable for at least 15 days [149].

Tests with oregano EO on contaminated lettuce detected, on leaves submerged in 0.6 mg/mL oregano oil, a 2 log CFU/g reduction in *L. monocytogenes* count after a 10 min treatment [173]. More drastic antimicrobial effects could be observed when oregano (0.125%) was directly mixed into meatballs, as, in this case, a 7 log decrease in CFU was detected in 1 h. This method was applied in sous-vide-processed salmon, and the effectiveness was enhanced by heating the samples to 55 °C; thus, a 7 log CFU/g decrease could already be achieved after 15 min [108]. With a higher concentration of oregano and/or temperature, the effects were even more pronounced.

According to the results of previous in vitro studies, cinnamon also possesses a characteristic antilisterial effect [67], which was confirmed in an in vivo study comparing it with thyme and oregano on the surface of radish sprouts. Cinnamon caused a nearly 1.5 log reduction in *L. monocytogenes* CFU on radish sprouts, after it was exposed for 24 h in a gaseous phase in a closed miniature jar system with an inner headspace of 1 L [159]. Differences were revealed on the basis of the applied EO volume filled in the jar. In contrast to others, this series of experiments was performed at 30 °C and gave clear-cut evidence of the differences in the antilisterial effects of cinnamon, thyme, and clove when applied in the same system, proving the usefulness of the gaseous phase method.

In another study, the effect of cinnamon and thyme was compared when they were applied by mixing them into the food matrix, more precisely into beef meat balls [44]. In this system, there were spectacular differences between these two EOs, with cinnamon showing a more moderate effect than thyme. With a 7.5% cinnamon concentration, a 7 log CFU/g drop could be reached in 1 h, despite the fact that the 5% concentration was already ineffective. In the case of thyme, however, these key concentration values inside the meatballs were 0.25% and 0.125%, respectively [44].

Cinnamon was also tested on the surface of smoked salmon, where a temperature- and strain-dependent effect was reported. In these surface applications, cinnamon was only able to inhibit the growth of *L. monocytogenes* strains at 4 °C. If experiments were performed at 10 °C, a constant proliferation was seen, with a strain-dependent kinetic pattern [152]. This study highlighted a practical challenge; that is, in the case of *L. monocytogenes,* EOs have strain-dependent effects [44].

The temperature dependence of the antilisterial effect of cinnamon was investigated in vanilla cream. Although, at 4 °C, the inhibitory effect on proliferation was unambiguous, it could not be observed at higher temperatures, such as 8 °C, 12 °C, and 16 °C [156]. The described temperature-dependent antilisterial effect was consistent with another previous report [152].

As in the previous case, a slight inhibitory effect could be observed if cinnamon was applied on chicken meat surfaces, integrated into sodium alginate coating [170] and polylactide film [148]. In both cases, a nearly 0.7 log CFU/g drop was reached compared to the CFU value of the control sample (5.5 log) in an experimental setup of more than 2 weeks at 4 °C.

The strain-dependent effects of EOs can be improved if EO combinations are used in one system. Recently, the combined antilisterial effect of EOs, such as rosemary and bay laurel (1% and 1.5%), and their nanoliposome-coated versions were tested at 4 °C in minced carp meat, revealing that their inhibitory effects were around 1 log CFU/g and 2 log CFU/g, respectively, in the investigated time range of 12 days [106]. Compared to other studies in which EOs were mixed into the matrix, these results were not outstanding, but drew attention to the fact that the effects of EOs could be improved using different coatings. 

Because of their easy application and biodegradability, the application of coatings is a preferred direction of research aiming to combat *L. monocytogenes* on foods. By applying EOs directly on surfaces, a ~1 log CFU/g decrease can be reached, although it certainly varies depending on the EO itself. For example, decontamination of surface-contaminated shrimp (8 log CFU/g) resulted in a 0.5 log CFU/g decrease if submerged into grape seed extract for 20 min [155]. This practice was also applied in a study in which lettuce was submerged in 0.7% safflower extract for 3 min, reaching a 1.58 log CFU/g reduction in *L. monocytogenes* count [174]. However, it was thought that the antilisterial effect could be further enhanced by an additional 1 log CFU decrease if EOs were applied with proper coatings. For that purpose, in vitro or in vivo tests were performed with different coating materials, such as wax [162,163], alginate-based [69], carboxymethyl cellulose-based [167], cellulose acetate [97], polyvinyl acetate [175] polylactide [148], chitosan-based [64,147,149,176], starch-based [177], sodium alginate [178], and gelatin-based [175,179] coatings. 

The successful usage of waxes on apples was described recently [162,163]. The application of a cardamon-based carboxymethyl cellulose nanoemulsion on the surface of tomato resulted in 3 log CFU/g less bacteria in 15 days compared to control. The starting CFU was 0.5 log CFU/g; while then the control CFU number increased to 4.5 log CFU/g, on the treated sample, a 1.5 log CFU/g *L. monocytogenes* was detected [180]. In another experiment, moringa EO was mixed into chitosan nanoparticles and gelatin, before being tested on a cheese surface. Starting with 3 log CFU/g *L. monocytogenes*, a 1 log CFU/g drop was seen at 4 °C in 10 days, and the bacterial proliferation could be inhibited even at 25 °C. The CFUs of controls reached around 6.5 log CFU/g and 8 log CFU/g at 4 °C and 10 °C, respectively [64]. A similar but somewhat less pronounced effect could be observed when chitosan nanofiber-based packaging material was tested with chrysanthemum on the surface of beef at 4 °C, 12 °C, and 25 °C for 7 days. At 4 °C and 12 °C, the CFU remained almost the same, while control CFUs increased by 3 and 3.5 log CFU/g, reaching 5.5 log CFU/g and 6 log CFU/g, respectively. At 25 °C, the restrictive antilisterial effect of chrysanthemum was still detectable, as the living cell count grew to 5.5 log CFU/g in the treated sample in contrast to the control, where this value was as high as 8.5 log CFU/g in 7 days.

The efficacy of cellulose acetate film was tested in a longer time period on sliced cheeses with pepper on 4 °C. The starting inocula were 4 log CFU/g, decreasing by the fourth day in the presence of pepper, but returning to the original level by day 24, while the CFU of the control firmly increased from 4 log CFU/g to 8 log CFU/g [97].

In the case of several foods, mixing EOs into the food matrix was shown to be a relevant method to hinder the proliferation of *L. monocytogenes*. Dannenberg et al. [97] tested the antilisterial effect of pepper EO not only on food surfaces, but also in food matrices in long-term experiments at 4 °C. During the 30 day experiment, they found that, when 2% pepper EO was present, the *Listeria* count increased from 4.5 log CFU/g to 5.5 log CFU/g, while the CFU in the control increased to 7 log CFU/g.

Other studies tried enhancing the antibacterial effect of pure EOs by presenting them in the form of nanoemulsions. In their comparative work, Elsherif et al. [96] mixed clove in a nanoemulsion into Egyptian Talaga cheese, detecting a stronger antilisterial effect of this EO. As a result, the starting 8 log CFU/g of *Listeria* in the so-prepared cheese dropped to 1.5 log CFU/g after 2–3 weeks, in contrast to the pure form of clove EO that could only achieve such an extent in reducing the CFU after 4–5 weeks at 4 °C. The applied EO concentrations were 0.78% and 1.56%, respectively.

The question of concentration is an important issue, as it should ideally harmonize with the taste characteristic of the food in or on which it is intended to be applied. Because of its refreshing flavor, mint is an ideal EO, the antilisterial effect of which was tested in mango and pineapple juices, adding concentrations of 0.625 mg/mL and 1.25 mg/mL, respectively [165]. A reduction in *L. monocytogenes* count from the starting 7 log CFU/mL value was observed, by 5 log CFU/mL in the case of pineapple, but only 1 log CFU/mL in the case of mango juice, at 4 °C in 15 min. The large differences between the antilisterial potentials could be due to the intrinsic characteristics of these juices, particularly their pH value and the potential antilisterial activity of pineapple.

The efficacy of mint was also tested in the form of coatings on the surface of fresh strawberries [167]. It was found that a nearly 2.5–3 log CFU/g drop could be achieved, compared to the control if carboxymethyl cellulose was the coating material; however, this could be further increased if chitosan was used for coating [167].

The efficacy of nanoemulsion-based coatings was demonstrated with lemongrass EO, showing that a nanoemulsion wax coating was able to inhibit *L. monocytogenes* proliferation, suppressing *L. monocytogenes* CFU/g by 4 log in 24 h of incubation at 37 °C [162]. Tea tree oil was proven to have a characteristic antibacterial effect on *L. monocytogenes* if applied in a 2% concentration to cucumber juice. At this concentration, the antilisterial effect was found to be independent of the tested temperatures, 4 °C and 25 °C [168], with effects already detected after 12 h. In the case of 0.25% concentration, however, no antilisterial effect was seen at 25 h and 48 h, although the killing effect at 4 °C was still 90% after 2 days. Tea tree EO proved to be very effective, drastically dropping the CFU in 20 min when applied to ground beef with a starting CFU of 5 log CFU/g [46]. This study focused on how the antilisterial effect of tea tree was influenced by the starting inoculum size of the bacterium (2 log, 3 log, 4 log, and 8 log CFU/g) in the meat matrix containing tea tree EO in 1.5% (*v*/*w*) concentration. The food matrix tests indicated that this EO had antimicrobial activity in all samples, except that in which the starting CFU was 8 log CFU/g.

Garlic EO in concentrations of 0.5% and 1.5% (*v*/*w*) was able to diminish the *Listeria* count in hummus even in the case of a very high (6 log CFU/g) starting CFU [145]. More precisely, garlic EO at 1.0–3.0% reduced *L. monocytogenes* count at day 10 by 0.7–3.0 and 1.3–3.6 log CFU/g at 4 and 10 °C, while the control CFU increased from 6 log CFU/g to 10 log CFU/g.

For cumin, a 0.2% concentration was enough to inhibit the proliferation of *L. monocytogenes* for 42 days in cheese, decreasing the living cell count from 3.8 log CFU/g to 3.2 log CFU/g, while the control bacterial count increased to 4 log CFU/g [181].

In a recent study, titration of the antilisterial effect of *Satureja horvatii* EO (a Greek-endemic plant) was performed in minced pork, and it was found that the antibacterial effect could already be observed at 0.6 mg/mL concentration, at both 4 °C (95%) and 25 °C (50%); this killing effect reached its maximum at both temperatures if the EO was applied in 10 mg/mL concentration [116]. These results were consistent with other studies, in which *Plectranthus amboinicus* EO was applied in beef patties at 4 °C for 15 days [73]. The authors found that, if the EO was used in 2 mg/mL concentration, the starting 5.7 log CFU/g decreased to 5 log CFU/g in 12 days, but increased to almost 6 log CFU/g by day 15. During this time, the control CFU permanently increased to 6.5 log CFU/g, similar to samples containing 1 mg/mL *Plectranthus amboinicus* EO.

A minced meat model was used to compare the antilisterial effect of *Lycium barbarum* (LBE) applied with and without chitosan coating in minced catfish at 4 °C for 14 days [153]. They found that, if LBE was applied together with chitosan, a more than 5 log CFU/g decrease could be observed in 14 days. From the results, it was also evident that chitosan itself hindered *L. monocytogenes*, but this antilisterial effect could be further improved by adding at least 0.4% LBE.

Nanoliposomes, due to their chemical nature, are also in focus for the food industry. This material was shown to increase the antilisterial effect of broccoli sprout extract (BSE) in ricotta cheese [91]. In this system, the control CFU increased from 3 log CFU/g to 4 log CFU/g in 12 days. If 0.8% free BSE was applied, the CFU dropped to 1 log CFU/g, whereas, when applied in a liposome, the CFU dropped to 0 in 9 days.

Salads have also been investigated as a food type in which EOs can be applied in the gaseous phase. Due to their loose texture, gases can easily reach inner surfaces, enabling EOs to exert their antilisterial effects. In a comparative study, fumes of different EOs, such as basil, carrot, cinnamon, clove, oregano, and thyme were tested on sprout, and their concentration-dependent antilisterial effect was tested at 30 °C in 24 h and ranked accordingly. For the experiments, 1250.0, 625.0, 312.5, 156.3, 78.1, 39.1, or 19.5 mg/L concentrations of EOs were used [159]. For oregano, thyme, and cinnamon, the MIC values were 78.1 mg/L and 156.3 mg/L, while, for carrot seed and basil, it proved to be 625 mg/L.

The antilisterial effect of lime was investigated in a very recent study in ready-to-eat salads at 4 °C in a 24 h test, performed in 1 L volume paper bags [104]. A volume of 200 µL caused a 3 log reduction compared to the control in 7 days. Application of 50 µL of lime EO still produced a 0.3 log CFU/g reduction. 

Integration of *Pimenta dioica* EO containing β-cyclodextrin complex and its gaseous-phase application on 25 g of ready-to-eat salads at 7 °C for 6 days revealed that the system did not have any significant in vivo effect on the salad, although it proved to be antilisterial when tested in vitro on agar plates [161]. 

There is a special group of foods where the antibacterial and antilisterial effects of EOs could be particularly useful. This is the so-called sous-vide technology that we previously described in relationship to salmon and oregano [108]. Application of oregano drastically lowered the viable *L. monocytogenes* count, decreasing the living cell count at 55.5 °C by 4 log CFU/g in 50 min in minced salmon. This effect could be enhanced by applying higher temperatures and longer incubation times. 

Weaker results were detected when salvia was applied in sous-vide cooked beef after it was chilled to 2–8 °C for 28 days [115]. In this case, a 1 log CFU/g drop was detected, compared to control. The authors suggested to use additional treatments in order to obtain satisfactory results. In another study, the authors presented better results in a sous-vide system by applying thyme and rosemary [144]. They mixed 0.1 mL of EO into 10 g of beef; then, after vacuum packaging, heat treatment was carried out, and the food was chilled to 2 °C and 8 °C. CFU determination after 28 days revealed definitively better results already in the case of controls, in which a 1 log CFU/g drop was detected at 2 °C in contrast to 4 °C, where this value was 8.3 log CFU/g. At 2 °C, rosemary and thyme were able to diminish the CFUs to 2.94 and 3.6 log CFU/g, while, in the case of 8 °C, the CFU values reached 5.67 and 8.24 CFU/g, respectively.

### 8.3. Antilisterial Effects of EO Components Tested in Different Food Systems

EOs are complex mixtures of compounds that can have either synergistic or antagonistic effects. Since the major components are mostly responsible for the biological activities, some recent studies concentrated on their antilisterial effects in pure forms. The antilisterial effects of camphor, verbenone [158], thymol, carvacrol, cinnamaldehyde [182], citral [164], eugenol, and vanillin [150,169] were the focus of several studies in various food systems.

In a comparative study, the antilisterial effect of thyme and its two compounds, verbenone and camphor, was analyzed in a modified gas atmosphere on fresh-cut lettuce, melon, and pineapple at 4 °C [158]. Treatments were performed by dipping the contaminated samples in a solution, containing 150 mg/L EO or compounds. Thyme proved to have superior activity in the case of lettuce, where the CFU was reduced from 2 log CFU/g to 0. The effect of verbenone was very similar in its tendency, while camphor did not have any significant antilisterial effect [158].

A similar dipping system was applied by Osaili and coworkers, who investigated the antilisterial activity of thymol, carvacrol, cinnamaldehyde, eugenol, and vanillin in marinated camel meat chunks at 4 °C and at 10 °C [150,182]. They found that eugenol lowered the CFU of *L. monocytogenes* by 1.9 log CFU/g, whereas vanillin achieved a reduction by 1.3 log CFU/g when these compounds were present in a concentration of 1% or 2%. The antimicrobial efficacy of a compound either alone or combined with marinade was higher at 10 °C than at 4 °C [150]. In contrast, the same concentrations of thymol, carvacrol, and cinnamaldehyde proved to be ineffective in decreasing the *L. monocytogenes* count at 4 °C or 10 °C [182].

Citral, in the form of nanoemulsions (0.15 and 0.3 mg/mL), had the capacity to decrease the *L. monocytogenes* count from 6 log CFU/g to 1 CFU/g on surfaces of fresh-cut papaya and melon in 60 h at all investigated temperatures (4 °C, 8 °C, 12 °C, and 16 °C) [164].
microorganisms-11-01364-t002_Table 2Table 2Antilisterial efficacies of essential oils and some EO compounds investigated in and on different foods. CFU values are rounded values and represent the ranges. Some studies used different EO concentrations and temperatures; here, we represent only the best combinations of these conditions. For further details, readers are referred to the original articles.FoodMethodApplied EO*L. monocytogenes* CFU before Treatment*L. monocytogenes*CFU after Treatment (Control)Applied TemperatureTime of IncubationReferenceVegetablesShredded and frozen*Thymus vulgaris*10^4^/g10^2^/g(10^4^/g)−22 °C14 daysZakrzewski, Purkiewicz et al. [98]Beef slicesSurface*Punica granatum* with chitosan–starch film coating10^5^/g10^8^–10^9^/g(10^10^/g)4 °C21 daysMehdizadeh, Tajik et al. [147]Beef minceSpray-drying*Thymus vulgaris*10^4^/g10^2^/g4 °C14 daysRadünz, dos Santos Hackbart et al. [172]Vegetable and fruit slicesMAP*Thymus vulgaris* and verbenone10^2^/g0/g(10^3^/g)4 °C7 daysScollard, Francis et al. [158]Radish sproutsGaseous*Thymus vulgaris**Orignaum vulgare, Cinnamon zeylanicum*10^6^/g10^6^/g10^6^/gin all cases 10^3^/g (10^5^/g)30 °C24 hLee, Kim et al. [159]Pork filletsSurface*Orignaum vulgare* with and without chitosan film10^4^/g10^2^/g (10^5^/g)4 °C15 daysPaparella, Mazzarrino et al. [183]Leafy vegetables and vegetable mediumMixed*Orignaum vulgare**Rosmarinus officialis*10^7^/g10^5^/g (10^7^/g)28 °C10 minde Medeiros Barbosa, da Costa Medeiros et al. [173]Salmon sous-videSurface*Orignaum vulgare*10^8^/g10^1^/g (10^5^/g)55–62.5 °C15 minDogruyol Mol et al. [108]Beef meatballsSparged*Orignaum vulgare**Rosmarinus officialis**Thymus vulgaris**Cinnamon zeylanicum*10^8^/g10^8^/g10^8^/g10^8^/gStrongly concentration-dependent (10^12^/g)4 °C1–72 hoursPesavento, Calonico et al. [44]Smoked salmonSurface*Cinnamomum javanicum*10^4^/gStrain- and temperature-dependent effect4 and 10 °C30 daysYuan, Lee et al. [152]Vanilla creamSparged*Cinnamonum zeylanicium*10^3^/g10^5^/g and 10^1^/g at 4°C (10^8^/g),above that ineffective4, 8, 12, 16 °C30 daysLianou, Moschonas et al. [156]Chicken meatSurfaceCinnamon oil (Ceylon type) with and without polylactide film and PEG10^6^/g10^5^/g (10^6^/g)Applied pressure increased the effect4 °C16 daysAhmed, Hiremath et al. [148]CarpMinced*Laurus nobilis**Salvia rosmarinus*10^4^/g10^7^–10^9^/g (10^10^/g)4 °C12 daysAala, Ahmadi et al. [106]CheeseSurface*Moringa oleifera*/chitosan10^3^/g10^2^/g (10^6^/g)4 and 25 °C10 daysLin, Gu et al. [64]CheeseSparged*Schinus terebinthifolius* Raddi10^4^/g10^5^/g(10^7^/g)4 °C30 daysDannenberg, Funck et al. [97]Pineapple and mango juiceMixed*Mentha arvensis* and *M. piperita*10^7^/mL10^2^/g PAEO and 10^6^/g MEO10^7^/mL4 °C15 minGuedes de Souza et al. [165]Cucumber juiceMixed*Melaleuca alternifolia*10^5^/mL0/g(10^5^/g)4 and 25 °C48 hShi, Zhang et al. [168]Watermelon juiceMixed*Melissa officinalis*10^6^/mL10^1^/mL (10^7^/g)4 °C7 daysCarvalho, Coimbra [74]StrawberrySurface*Mentha spicata*/carboxymethyl cellulose and chitosan10^2^/g10^3^–10^5^/g(10^7^/g)4 °C12 daysShahbazi [167]AppleSurface*Cymbopogon citratus*/wax10^8/g^0/g(10^8^/g)37 °C24 hJo, Song et al. [162]Ground beefSparged*Melaleuca alternifolia*10^2^–10^8/g^Depended on the starting CFU4 °C14 daysSilva, Figueiredo et al. [46]HummusSparged*Allium sativum*10^6/g^10^3^/g(10^10^/g)4 and 10 °C10 daysOlaimat, Al-Holy et al. [145]CheeseSparged*Cuminum cyminum*10^3^/g10^3/g^(10^4^/g)4 °C42 daysHassanien, Mahgoub et al. [181]Pork meatMinced*Satureja horvatii*10^6^/g0/g(n.d.)4 and 25 °C72 hBukvicki, Stojkovic et al. [116]Beef pattiesMinced*Plectranthus amboinicus*10^6^/g10^6^/g(10^7^/g)7 °C15 daysDutra da Silva, Bernardes et al. [73]CatfishMinced*Lycium barbarum*/chitosan10^4^/g10^1^/g(10^7^/g)4 °C14 daysAlsaggaf, Moussa et al. [153]SaladGaseous*Citrus aurantiifolia*10^3/g^10^6^/g(10^8^/g)4 °C7 daysParichanon, Sattayakhom et al. [104]SaladGaseous*Pimenta dioica*/β-cyclodextrin10^7^/g10^5^/g(10^7^/g)6 °C7 daysMarques, Grillo et al. [161]Salad (Tzatziki)Gaseous*Citrus limon*10^6/g^10^1^/g(10^4^/g)4, 10, 21 °C70 daysTsiraki, Yehia et al. [160]Beef, sous-videSurface*Salvia officinalis*10^5^/g10^4^/g(10^5^/g)2–8 °C28 daysMoura-Alves, Gouveia et al. [115]Beef, sous-videMinced*Rosmarinus officialis**Thymus vulgaris*10^5^/g10^5^/g10^3^/g10^4^/g(10^8^/g)2–8 °C28 daysGouveia, Alves et al. [144]Italian salamiSparged*Origanum vulgare*-*Cinnamon cassia* with sodium alginate10^4^/g10^4^/g(10^5^/g)4 °C7 daysGottardo, Biduski et al. [86]Turkey filletSurface*Trachyspermum ammi* with alginate coating10^6/g^10^5^/g(10^9^/g)4 °C12 daysKazemeini, Azizian et al. [94]Ricotta cheeseSpargedBroccoli sprout with liposome10^3^/g0–10^2^/g(10^4^/g)4 °C12 daysAzarashkan, Farahani et al. [91]Ricotta salad cheeseGaseous*Citrus limon* var. *pompia*10^5^/g10^3^–10^5^/g(10^6^/g)5 °C30 daysFancello, Petretto et al. [92]CheeseSparged*Cymbopogon citratus* entrapped in liposome10^3^/g10^3^–10^6^/g(10^7^/g)4 and 25 °C14 daysCui, Wu et al. [95]Talaga cheeseSparged*Cuminum cyminum*Cinnamon10^8^/g0–10^5^/g(10^7^/g)4 °C3 weeksElsherif and Talaat Al Shrief [96]Cheese mimicking modelSparged*Cinnamon cassia**Thymus vulgaris*10^6^/g10^3^/g(10^6^/g)10 °C24 daysde Carvallo, de Souza et al. [88]ShrimpSurfaceGrape seed extract with and without nisin10^8^/g10^7^/g(10^8^/g)Only treatmentNo incubationZhao, Chen et al. [155]ShrimpSurface*Alpinia galangal, Rosmarinus officinalis, Eucalyptus staigerana*10^8^/g10^8^/g(10^9^/g)4 and 8 °C16 daysWeerakkody, Caffin et al. [82]LettuceSurface*Carthamus tinctorius*10^5^/g10^6/g^(10^6^/g)4 °C8 daysSon, Kang et al. [174]Tomato juiceMixed*Cinnamon cassia*, *Cymbopogon citratus**Thymus vulgaris*10^5^/g10^3^–10^4^/g(10^6^/g)10 and 25 °C48 hKim, Kim et al. [81]BeefMinced*Ceratonia siliqua*10^2^/g0/g(10^6^/g)7 °C10 daysHsouna, Trigui et al. [65]Hotdog sausageSurface*Thymus vulgaris**Syzgium aromaticum*10^5^/g10^4^/g(10^5^/g)4 °C12 hSingh, Singh et al. [83]WatermelonSurface*Origanum vulgare*10^5^/g10^1^–10^4^/g(10^5^/g)4 °C3 daysZhu, Wei et al. [166]

EO compounds




Milk, bovineMixedEugenol nanodispersion10^5^/mL0–10^7^(10^8^/g)21 °C50 hShah, Davidson et al. [169]Papaya and melonSurfaceCitral nanoemulsion10^5^/g10^1^–10^4^/g(10^5^/g)4, 8, 12, 16 °C180 hLuciano, Pimentel et al. [164]Camel meat, marinated and unmarinatedSpargedCarvacrol, thymol, cinnamaldehyde10^5^/g10^1^–10^4^/g(10^5^/g)4 and 10 °C7 daysOsaili, Hasan et al. [182]Camel meat, marinated and unmarinatedSpargedEugenol, vanillin10^5^/g10^5^/g(10^6^/g)4 and 10 °C2 daysOsaili, Al-Nabulsi et al. [150]


## 9. Conclusions

From the above studies, it is evident that certain EOs administered alone have the capacity to deter the presence of *L. monocytogenes* in food. However, several EOs have a strain-dependent antilisterial effect, which has to be considered for applications in the food industry. From a methodological point of view, the most successful approaches are those, independent of the food type, in which EOs with antilisterial effects were mixed into the food matrix, such as cheese, salad, mixed fish, and minced meat in form of meatballs or sausage. Their efficacy could be further increased if nanoemulsions were applied. Research also revealed that gaseous-phase EO-based treatments are reliable methods for practical applications, especially in the case of salads. Another conclusion was that temperature can influence the efficacy of EOs in or on the food matrices. Furthermore, coatings, especially biodegradable coatings, are forward-looking applications, since a 1 or 2 log reduction in CFU could be achieved using this technology compared to the noncoated formulations of the same EOs. The typically applied concentrations of EO or their compounds were around 0.5–2%, already influencing the flavor characteristics of food. Therefore, the applied EO should harmonize with the food type to be treated, e.g., mint with fruit juices, cinnamon with vanilla cream, and thyme or rosemary with beef.

## Data Availability

No data is available concerning to this manuscript.

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
