# Peer review of "Potential of Essential Oils in the Control of Listeria monocytogenes"

_microorganisms, 2023, doi:10.3390/microorganisms11061364_

Round 1

Reviewer 1 Report

The work entitled “Potential of Essential Oils in the control of Listeria monocytogenes” is interesting and useful, but some issues are necessary to address for the manuscript to be suitable for publication. Throughout the manuscript, the digits must be separated by a point, and overlapping words must be avoided. 

Line 25:  “Food matrices” instead of “Foodmatrixes”;

line 27:  “Antilisterial” instead of “Antiistrial “;

line 31: “Food industry and food preservatives” instead of “Foodindustry and foodpreserrvatives”;

line 31: You must to correct this word “raison d’etre”;

line 38: The case fatality rate of listeriosis worldwide must be added;

line 60: The value of low water activities must be added;

line 125: This information is not correct “EOs are complex mixtures, which can consist of approximately 20-60 components”, since the EOs contain hundreds of components. You must correct it;

line 304: It would be better to add a column containing the MIC and MBC values in Table 1;

line 416: In order to properly present and facilitate the reading of the results, a Table containing (EOs, food treated with L. monocytogenes, concentration of EOs used, temperature and time of incubation, density of Listeria monocytogenes before and after treatment) should be drawn up.

Furthermore, Table 1 must be discussed in the text.

The English language should be improved

Author Response

Detailed Responses to Editor and Reviewers

Manuscript ID: microorganisms-2349781

Response to the Editor’s comments

We are very pleased to resubmit for publication the revised version of  “Potential of Essential Oils in the control of Listeria monocytogenes” by György Schneider, Anita Steinbach, Ákos Putics, Ágnes Solti-Hodován, Tamás Palkovics for consideration for publication in MDPI-Microorganisms in the special issue “An update on Listeria monocytogenes” to be considered for publication as an review article in MDPI Microorganisms. We are grateful for your positive opinion about our manuscript. We considered your comments hoping our revision has improved the paper to a level of your satisfaction.

Here we present your comments / answers in red, while we give our response in black.

Response to Reviewer #1 Comments

Reviewer 1

The work entitled “Potential of Essential Oils in the control of Listeria monocytogenes” is interesting and useful, but some issues are necessary to address for the manuscript to be suitable for publication. Throughout the manuscript, the digits must be separated by a point, and overlapping words must be avoided. 

Thank You for your positive comment about our work. One critic of You was the quality of English. So we have improved it throughout the text. The suggested corrections listed bellow were performed  

Line 25:  “Food matrices” instead of “Foodmatrixes”;

Corrected. (Line 27)

line 27:  “Antilisterial” instead of “Antiistrial “;

Corrected. (Line 27)

line 31: “Food industry and food preservatives” instead of “Foodindustry and foodpreserrvatives”;

Corrected. (Line 32)

line 31: You must to correct this word “raison d’etre”;

Corrected by reformulating the sentence. (Line 32)

line 38: The case fatality rate of listeriosis worldwide must be added;

The data is added. (20%) (Line 39)

line 60: The value of low water activities must be added;

No precise data was found for that.

line 125: This information is not correct “EOs are complex mixtures, which can consist of approximately 20-60 components”, since the EOs contain hundreds of components. You must correct it;

Correceted: The number of identified components usually ranges from 100 to 250, but in some oils (lavender, geranium, rosemary) 450 to 500 chemicals have been found. (Line 126)

line 304: It would be better to add a column containing the MIC and MBC values in Table 1;

Yes it is reasonable, and we added two extra columns containing these data.

line 416: In order to properly present and facilitate the reading of the results, a Table containing (EOs, food treated with L. monocytogenes, concentration of EOs used, temperature and time of incubation, density of Listeria monocytogenes before and after treatment) should be drawn up.

Summary of the food based EO experiments are now also summarized in form of a table as Table 2.

Furthermore, Table 1 must be discussed in the text.

Before Table 1. there is a nearly one page description. We did not want to go in detail concerning to the results, but in this text section our aims were to flash up some relevant aspects with that the antilisterial activities of certain EOs were investigated.

Reviewer 2 Report

"Potential of Essential Oils in the control of Listeria monocytogenes" very good work. All the nuances are judged, the authors have worked hard. For the food industry, this is already applicable and can be developed further. However, in practice, microorganisms are rarely present on the substrate as single cells; they most often form biofilms where they combine with other microorganisms. In the future, we would like to recommend the authors to write a similar review for biofilm microorganisms

Author Response

Detailed Responses to Editor and Reviewers

Manuscript ID: microorganisms-2349781

Response to the Editor’s comments

We are very pleased to resubmit for publication the revised version of  “Potential of Essential Oils in the control of Listeria monocytogenes” by György Schneider, Anita Steinbach, Ákos Putics, Ágnes Solti-Hodován, Tamás Palkovics for consideration for publication in MDPI-Microorganisms in the special issue “An update on Listeria monocytogenes” to be considered for publication as an review article in MDPI Microorganisms. We are grateful for your positive opinion about our manuscript. We considered your comments hoping our revision has improved the paper to a level of your satisfaction.

Here we present your comments / answers in red, while we give our response in black.

Response to Reviewer #2 Comments

"Potential of Essential Oils in the control of Listeria monocytogenes" very good work. All the nuances are judged, the authors have worked hard. For the food industry, this is already applicable and can be developed further. However, in practice, microorganisms are rarely present on the substrate as single cells; they most often form biofilms where they combine with other microorganisms. In the future, we would like to recommend the authors to write a similar review for biofilm microorganisms.

Authors thank the positive opinion about our work for Reviewer2. Yes sure biofilm especially on food matrices is an important issue.

Round 2

Reviewer 1 Report

Dear authors,

the manuscript has been improved but some other modifications are needed. 

1) In some cases, the explanation of the abbreviations lacks: for example, in line 42 there isn't for ECDC and EFSA; in line 133 there isn't for FDA. The same for TEMPO, DEM, and VIDAS in line 250.

2) In lines 118-120 there is a repetition of issues already discussed before.

3) Standardize the abbreviation of "Essential oil": in some cases, you reported, "EOs" while in others "Eos" (line 332).

4) Once abbreviated it's needless to repeat the full explanation; I mean "Essential oil" that should be abbreviated also in lines 165, 172, 223, 289, 290, 398, 485, 875.

5) The explanation for "MIC" and "MIB" has been reported before: don't repeat it in line 322. 

6) Table 1 should be checked carefully: the column "Publication" could be replaced with a row titled "Reference" as in Table 2. Furthermore, it's enough to indicate only the number of the reference. There are some abbreviations without explanation; so, I suggest adding a list of abbreviations at the end of the manuscript. There are some mistakes in the names of compounds of Eugenia sp., Origanum vulgare subs. hirtum, and thymus capitatus. The column "Experimental condition" content should be reorganized and you should choose what insert: the experimental conditions or the food sources. If you want to talk about both, you should create separate columns. 

7) Line 478 correct "marjoram".

8) In Table 2, check the column of temperature for the watermelon row.

9) As I just said in the previous report, you should standardize the decimal separators: in some cases, you used a point while in others a comma.

 Minor editing of English language required

Author Response

Answers to the comments of Reviewer 2.

(2nd review process)

Dear authors,

the manuscript has been improved but some other modifications are needed. 

  • In some cases, the explanation of the abbreviations lacks: for example, in line 42 there isn't for ECDC and EFSA; in line 133 there isn't for FDA. The same for TEMPO, DEM, and VIDAS in line 250.

Thnak you it is reasonable so abbrevations are explained now.

  • In lines 118-120 there is a repetition of issues already discussed before.

Corrected

  • Standardize the abbreviation of "Essential oil": in some cases, you reported, "EOs" while in others "Eos" (line 332).

EO is the accepted one. Now „Eos” is corrected to „EOs” throughout the text. (Line 251)

  • Once abbreviated it's needless to repeat the full explanation; I mean "Essential oil" that should be abbreviated also in lines 165, 172, 223, 289, 290, 398, 485, 875.

Thank You for your comment. We have corrected them. We left the „essential oil” as a complete word in subtitles and table legends.

  • The explanation for "MIC" and "MIB" has been reported before: don't repeat it in line 322. 

It is corrected in this version.

  • Table 1 should be checked carefully: the column "Publication" could be replaced with a row titled "Reference" as in Table 2. Furthermore, it's enough to indicate only the number of the reference. There are some abbreviations without explanation; so, I suggest adding a list of abbreviations at the end of the manuscript. There are some mistakes in the names of compounds of Eugenia sp., Origanum vulgare subs. hirtum, and thymus capitatus. The column "Experimental condition" content should be reorganized and you should choose what insert: the experimental conditions or the food sources. If you want to talk about both, you should create separate columns. 
  • Table 1: „Publication” is corrected to „Reference”  
  • Thank You for your remark, but adding the authors was intentional as w efound it more „personal”.
  • The abbreviation list is added at the end of the text, now.
  • Names of the EOs are corrceted recently, like:

Eugenia sp --> Eugenia spp.

Origanum vulgare subs. hirtum --> Origanum vulgare subsp. hirtum

thymus capitatus --> for us it seems to be OK.

  • Concerning to your remearl it is reasonable so experimental food sources and conditions were separated.

7)    Line 478 correct "marjoram".

Corrected.

8) In Table 2, check the column of temperature for the watermelon row.

Thank You, it was originally omitted. No this information is inserted: 4 °C.

9) As I just said in the previous report, you should standardize the decimal separators: in some cases, you used a point while in others a comma.

Yes sure, there were some things to correct, but now they are OK.
